# First Stabilize and then Gradually Recruit: A Paradigm Shift in Protective Mechanical Ventilation for Acute Lung Injury

**DOI:** 10.3390/jcm12144633

**Published:** 2023-07-12

**Authors:** Gary F. Nieman, David W. Kaczka, Penny L. Andrews, Auyon Ghosh, Hassan Al-Khalisy, Luigi Camporota, Joshua Satalin, Jacob Herrmann, Nader M. Habashi

**Affiliations:** 1Department of Surgery, Upstate Medical University, Syracuse, NY 13210, USA; niemang@upstate.edu; 2Departments of Anesthesia, Radiology and Biomedical Engineering, University of Iowa, Iowa City, IA 52242, USA; 3Department of Medicine, R Adams Cowley Shock Trauma Center, University of Maryland Medical Center, Baltimore, MD 21201, USA; 4Department of Medicine, Upstate Medical University, Syracuse, NY 13210, USA; 5Brody School of Medicine, Department of Internal Medicine, East Carolina University, Greenville, NC 27834, USA; 6Department of Adult Critical Care, Guy’s and St Thomas’ NHS Foundation Trust, King’s Partners, St Thomas’ Hospital, London SE1 7EH, UK; 7Department of Biomedical Engineering, University of Iowa, Iowa City, IA 52242, USA

**Keywords:** acute respiratory distress syndrome, ARDS, ventilator-induced lung injury, VILI, protective mechanical ventilation, open lung approach, OLA, stabilize lung approach, SLA, time-controlled adaptive ventilation, TCAV, airway pressure release ventilation, APRV

## Abstract

Acute respiratory distress syndrome (ARDS) is associated with a heterogeneous pattern of injury throughout the lung parenchyma that alters regional alveolar opening and collapse time constants. Such heterogeneity leads to atelectasis and repetitive alveolar collapse and expansion (RACE). The net effect is a progressive loss of lung volume with secondary ventilator-induced lung injury (VILI). Previous concepts of ARDS pathophysiology envisioned a two-compartment system: a small amount of normally aerated lung tissue in the non-dependent regions (termed “baby lung”); and a collapsed and edematous tissue in dependent regions. Based on such compartmentalization, two protective ventilation strategies have been developed: (1) a “protective lung approach” (PLA), designed to reduce overdistension in the remaining aerated compartment using a low tidal volume; and (2) an “open lung approach” (OLA), which first attempts to open the collapsed lung tissue over a short time frame (seconds or minutes) with an initial recruitment maneuver, and then stabilize newly recruited tissue using titrated positive end-expiratory pressure (PEEP). A more recent understanding of ARDS pathophysiology identifies regional alveolar instability and collapse (i.e., hidden micro-atelectasis) in both lung compartments as a primary VILI mechanism. Based on this understanding, we propose an alternative strategy to ventilating the injured lung, which we term a “stabilize lung approach” (SLA). The SLA is designed to immediately stabilize the lung and reduce RACE while gradually reopening collapsed tissue over hours or days. At the core of SLA is time-controlled adaptive ventilation (TCAV), a method to adjust the parameters of the airway pressure release ventilation (APRV) modality. Since the acutely injured lung at any given airway pressure requires more time for alveolar recruitment and less time for alveolar collapse, SLA adjusts inspiratory and expiratory durations and inflation pressure levels. The TCAV method SLA reverses the open first and stabilize second OLA method by: (i) immediately stabilizing lung tissue using a very brief exhalation time (≤0.5 s), so that alveoli simply do not have sufficient time to collapse. The exhalation duration is personalized and adaptive to individual respiratory mechanical properties (i.e., elastic recoil); and (ii) gradually recruiting collapsed lung tissue using an inflate and brake ratchet combined with an extended inspiratory duration (4–6 s) method. Translational animal studies, clinical statistical analysis, and case reports support the use of TCAV as an efficacious lung protective strategy.

## 1. Introduction

Acute respiratory distress syndrome (ARDS) remains a significant clinical problem, with the primary treatment being supportive with positive pressure mechanical ventilation [1]. Unfortunately, mechanical ventilation is associated with unintended ventilator-induced lung injury (VILI), which can significantly increase mortality [2] despite current protective ventilation strategies [3,4,5,6]. Pulmonary edema and surfactant deactivation alter the inflation and deflation mechanics of alveoli and alveolar ducts (Figure 1) [7,8,9,10,11]. Early evidence based on computerized tomographic (CT) imaging characterized ARDS by two distinct anatomic and functional lung compartments: (1) a dependent compartment of nonaerated tissue arising from edema, surfactant dysfunction, and atelectasis; and (2) a non-dependent aerated compartment with smaller volume compared with a normal adult lung (i.e., the “baby lung”) [12]. Such understanding led to the ARDSnet “protective lung approach” (PLA) using low tidal volumes and plateau pressure (Pplat) to protect the baby lung from overdistension (i.e., volutrauma). Marini and Gattinoni recently described the role of mechanical ventilation in the pathogenesis of VILI as a positive feedback mechanism, whereby overdistension of the baby lung leads to localized injury and subsequent loss of aeration, resulting in a shrinking baby lung with higher susceptibility to further overdistension [13]. The progressive loss of aerated tissue due to lung collapse is associated with an increased risk for VILI, which pushes the lung into a “VILI Vortex”. Attempts have been made to prevent the lung from entering the VILI Vortex using an “open lung approach” (OLA) via the application of high positive end-expiratory pressure (PEEP), with or without recruitment maneuvers (RMs). Despite the PLA and OLA lung protective strategies, the mortality associated with ARDS has remained essentially unchanged for two decades [14,15,16,17].

The PLA used volume assist control mode in the original ARDS Network (ARDSnet) randomized controlled trial (RCT) (Figure 2A), which soon became a standard-of-care for mechanical ventilation patients with ARDS [27]. Collapsed tissue in the dependent compartment significantly reduces the volume of functional lung tissue available for gas exchange and increases parenchymal strain, which may initiate the VILI Vortex [13,28]. Strain is defined as the size change of a structure to an applied load. Thus, the change in the size of alveoli and alveolar ducts with the applied force of mechanically delivered tidal volume (V_T_) is the parenchymal strain. The conceptual goals of a PLA are to protect aerated tissue from overdistension (i.e., volutrauma), while applying sufficient PEEP to prevent progressive lung collapse [27]. The bulk of the atelectatic tissue is effectively allowed to “rest”, remaining collapsed throughout the ventilatory cycle [27]. Although the original ARDSnet trial demonstrated a significant reduction in mortality for patients ventilated with a tidal volume (V_T_) of 6 mL kg^−1^ compared with those ventilated with 12 mL kg^−1^ [27], subsequent analyses suggested that a fixed, limited V_T_ was not always protective, but rather that outcomes were better predicted by the ratio of V_T_ to lung compliance [29].

Deans et al. analyzed the outcomes in 2587 patients that were excluded from the original ARDSnet trial for procedural reasons but were treated with the standard-of-care at the time (i.e., V_T_ ≅ 10 mL kg^−1^) [29]. The group receiving standard-of-care ventilation had similar mortality to patients receiving low V_T_. Moreover, high V_T_ increased mortality in patients with low respiratory system compliance (C_RS_), yet reduced mortality in those with high C_RS_ [29]. Thus, whether ventilation is protective or injurious depends on not only the size of V_T_, but also C_RS_ and the end-expiratory lung volume (EELV) into which V_T_ is being delivered.

Hence, Driving Pressure (∆P) has been proposed to be a more robust indicator of the impact of V_T_ for a given C_RS_ (i.e., ∆P = V_T_/C_RS_). Moreover, ∆P is better at stratifying the mortality risk of ARDS than V_T_ alone [4,31,32,33,34,35]. The combination of V_T_ and C_RS_ on mortality was studied in a retrospective analysis on 1096 ARDS patients using the Bayesian multivariable logistic regression [36], which showed that the impact of V_T_ on mortality was dependent on C_RS_, and suggested that ∆P should be used to guide protective lung ventilation [4,31,32,33,34,35,36]. Thus, EELV and lung pathophysiology (as reflected in C_RS_) are what predisposes the lung to mechanical damage during ventilation, rather than specific ventilator settings alone.

## 2. Conceptual Approaches to Protective Lung Ventilation

Although the PLA is designed to reduce overdistension of the baby lung and is the current standard-of-care in ARDS, the mortality associated with the syndrome remains high [3,4,5,6], as overdistension of the baby lung may not be the primary mechanism driving VILI. Previous studies have shown that normal aerated lung tissue is very resistant to high strain and overdistension-induced tissue damage [37,38,39,40,41,42,43,44,45]. Moreover, the presence of repetitive alveolar collapse and expansion (RACE) [46] and micro-atelectasis results in stress concentrators scattered throughout the lung (Appendix A) [8,9,10,47,48], which cannot be seen with conventional chest radiography or CT imaging [49]. These phenomena have been termed “hidden micro-atelectasis” [11], which, when present, may result in additional pathophysiologic complications. Indeed, it is well-known that persistent atelectasis is associated with multiple pathologies (Table 1).

### 2.1. Protective Lung Approach

The PLA is *constrained* to ventilating an injured lung with heterogeneously distributed regions of collapse since it does not attempt to reopen the lung. Such heterogeneity, combined with low V_T_, further contributes to the progressive loss of aerated volume and drives the VILI Vortex (Figure 3) [18]. Laffey et al. investigated potentially adjustable ventilator settings associated with ARDS mortality in 2377 patients from 50 countries enrolled in the LUNG SAFE study [4]. Ventilator settings associated with increased mortality were higher peak pressures, plateau pressures, PEEP, ∆P, and respiratory rate. By focusing on protocolized ventilator settings (i.e., V_T_ = 6 mL kg^−1^, Pplat < 30 cmH_2_O), rather than the unique pathophysiology and disease processes of a given patient, the clinician may fail to individualize appropriate therapy. For example, low V_T_ applied to a patient with high C_RS_ has been shown to increase mortality [29]. Parameters that take into account changes in lung physiology, such as Driving Pressure, have been shown to be better predictors of mortality.

### 2.2. Open Lung Approach

By contrast, the OLA is designed to reopen collapsed lung tissue using higher PEEP, with or without recruitment maneuvers (RMs) to release the constraints of ventilating a heterogeneous lung [14,15,16,17]. However, OLA failed to reduce ARDS-related mortality below that of the original ARDSnet low V_T_ strategy [27]. High-frequency oscillatory ventilation (HFOV) may also be viewed as an OLA strategy achieved by an entirely different ventilator method, but thus far has not shown superiority to the original ARDSnet protocol [65,66,67]. Several possible explanations have been proposed for these results, including (1) maintenance of an open lung with mechanical ventilation does not necessarily protect it from VILI. This is not likely since physiologic studies have shown that an open lung is highly resistant to injury during mechanical ventilation [68]; (2) the ventilation methods used to open the lung did so incompletely or transiently and, therefore, heterogeneity remained [69]; (3) high distending pressures during HFOV may result in hemodynamic compromise [70]; and (4) the maldistribution of oscillatory flow in a heterogeneously injured lung, resulting in regions of high parenchymal strain [71].

Thus, from multiple physiological perspectives, the efficacy of the LV_T_ PLA method is problematic. The outcome data support the physiologic-based assumptions, with many statistical and meta-analyses showing that the LV_T_ ventilation strategy has not further lowered mortality in patients with established ARDS [3,4,6,72,73,74,75] below that in the ARDSnet study published 23 years ago [27].

Data from previous OLA clinical trials support the hypothesis that the recruitment methods did not effectively open the lung, since there was no clear evidence of long-term lung recruitment based on lung imaging, arterial blood gases, or estimates of C_RS_. These negative trials have led some to suggest that the OLA should be abandoned, although the likely reason for failure was that the goals of the strategy were not achieved [76]. If regional RACE and micro-atelectasis result in stress concentrators that propagate VILI, then the most effective way to reduce VILI is to both reopen and stabilize the lung [11,37,40,41,47,48,77]. Danti et al. demonstrated that the combination of PEEP and RM duration plays a role in the mortality associated with ARDS. They found that: (1) higher PEEP without an RM was superior to lower PEEP; (2) higher PEEP with a prolonged RM was inferior to higher PEEP alone; and (3) higher PEEP with a brief RM was superior to higher PEEP alone. These data reveal the complex relationship between PEEP and RMs and are another explanation for why the OLA has not further reduced mortality in ARDS [78].

### 2.3. The Acutely Injured Lung Becomes Time- and Pressure-Dependent

ARDS causes the lung to become both *time*- and *pressure*-dependent. This means that it will take more time to recruit collapsed alveoli and less time for them to collapse at any given airway pressure. Changes in the time constants associated with alveolar opening and collapse result in greater susceptibility to progressive atelectasis and reductions in EELV, which decreases C_RS_ and increases ∆P for any given V_T_ [4,31,32,33,34,35]. The OLA attempts to eliminate atelectasis and normalize EELV; however, as discussed above, the OLA has not further reduced ARDS-related mortality [78]. Combined, these studies suggest that a better ventilation strategy must be developed to break the constraints of ventilating a heterogeneously injured lung (Figure 3) [64].

### 2.4. Stabilize Lung Approach

We postulate that the ventilation strategy necessary to break the constraints of ventilating a heterogeneously injured lung must (i) first stabilize the lung by halting RACE, since atelectrauma is a primary VILI mechanism, and then (ii) reopen the lung gradually so that it may heal in a more natural, inflated state (even though it is being supported by an unnatural mechanical ventilator) [18]. Unfortunately, identifying such a ventilation strategy has been elusive. If instability and collapse are the engines that drive the VILI Vortex, the goal of any protective ventilation strategy should be to eliminate instability as soon as possible. When atelectrauma is minimized, collapsed (but potentially recruitable) tissue may be reinflated slowly and safely over hours or even days. (Figure 3) [8,9,10,11].

Time-controlled adaptive ventilation (TCAV) is a strategy designed to stabilize lung tissue at risk for RACE, then gradually reopen collapsed tissue (Figure 3). TCAV utilizes a continuous positive airway pressure (CPAP) that is periodically interrupted by brief Release Phases (Figure 2B). The extended time at inspiration combined with an inflate and break ratchet method will gradually reopen collapsed tissue. The duration of this Release Phase is very brief and acts as a break to prevent newly recruited tissue from re-collapsing. The precise timing of the Release Phase necessary to prevent RACE is personalized and adaptive, based on evolving changes in C_RS_ [79,80]. TCAV is the most investigated method for setting the parameters of airway pressure release ventilation (APRV), and has been comprehensively studied and reviewed by multiple investigators (e.g., TCAVnetwork.org) [81,82,83,84,85,86,87,88]. TCAV is a personalized approach to ventilation and differs from other methods to set APRV parameters, which may not stabilize and open the lung as effectively [40,89]. The positive and negative attributes of the APRV mode are beyond the scope of this paper but have been addressed in detail in a recent publication [90].

## 3. Stabilize the Lung Approach

### 3.1. Time-Dependent Alveolar Collapse

The tendency for ongoing lung collapse is increased in ARDS [87,91,92]. In animal models of ARDS, loss of aeration during exhalation can occur as quickly as 0.6 s, with the majority of aeration loss occurring within the first 4 s of exhalation [93,94]. Thus, the expiratory duration of fewer than 0.6 s may prevent the collapse of alveoli with fast rates of closure [95]. In a porcine model of ARDS, 95% of aeration loss occurred within 0.8 s of expiration [94,96]. More specifically, poorly aerated regions of injured lungs also exhibited faster aeration loss during exhalation (i.e., shorter time constants) compared with normally aerated regions [94]. Lachmann et al. postulated that alveoli with multiple rates of collapse can be stabilized by dramatically shortening the expiratory time [97]. Accordingly, alveoli in the acutely injured lung can be stabilized by applying a very brief expiratory duration [91,93,94,96].

### 3.2. TCAV Method Stabilizes and then Opens the Lung

With TCAV, the brief expiratory duration does not allow time for the lung to fully de-pressurize, thus effectively maintaining a “time-controlled” PEEP (Figure 2B, TC-PEEP), which may vary regionally throughout the heterogeneous lung. The very brief expiratory duration generates a TC-PEEP and, combined, are highly effective in stabilizing the lung (Figure 2B, Release Phase) [82,83,84,85,86,87], pulling the lung from the VILI Vortex. [83,84,85,98]. More traditional approaches to ventilation (with longer expiratory durations) assume that alveoli are stabilized by pressure alone (PEEP), and thus allow ample time for alveolar collapse during exhalation (Figure 2A, Duration of Expiration) [99]. TCAV is also very efficient at reopening collapsed lungs by extending the CPAP Phase duration to recruit alveoli with long opening time constants and uses a very brief expiratory duration that acts as a brake to prevent re-collapse. Alveolar recruitment reestablishes alveolar interdependence and parenchymal tethering, which help minimize both RACE and stress concentrators pulling the lung from the VILI Vortex (Figure 3) [100,101,102,103,104].

### 3.3. Basic Scientific Evidence of TCAV Efficacy

In vivo microscopy of subpleural alveoli demonstrates that a brief expiratory time immediately stabilizes alveoli in animal models of acute lung injury, as compared with traditional applications of high PEEP (Figure 4). Moreover, if the Release Phase of TCAV is extended, the stabilization of alveoli is lost [105]. Roy et al. compared TCAV to conventional mechanical ventilation using traditional applications of PEEP, based on their relative efficacies to stabilize alveoli in a rodent model of ARDS induced by hemorrhagic shock, and demonstrated that TCAV was much more effective than high PEEP alone at stabilizing subpleural alveoli [98]. Moreover, alveolar stabilization was highly correlated with reduced injury based on histopathology, improved oxygenation at a similar peak inspiratory pressure, preservation of surfactant protein-B, and preservation of the alveolar/capillary barrier. Such lung protection was achieved even though the APRV group had a significantly higher V_T_ compared with the conventional group (12.9 ± 0.2 vs. 9.5 ± 0.2 mL kg^−1^) [98]. Although the results clearly show that TCAV was superior to high PEEP at stabilizing alveoli and reducing lung histopathology, these results were generated in a relatively short 6 h experiment.

Kollisch-Singule et al. showed that both high PEEP and TCAV significantly reduced alveolar micro-strain in vivo. In addition, TCAV significantly increased alveolar recruitment [83] and normalized alveolar size distribution, thus providing dynamic homogeneity similar to that of the control group [85]. Finally, TCAV maintained the micro-anatomic gas distribution in alveoli and alveolar ducts, similar to that seen in a normal lung, which was not attained using a high PEEP strategy alone [84]. A limitation of animal studies is that chest wall compliance is different from that of humans. However, even with this limitation, all studies support the efficacy of TCAV in reducing lung injury and death in clinically applicable animal models of sepsis and gut ischemia/reperfusion-induced ARDS and that the mechanism of lung protection is first stabilizing then gradually reopening collapsed lung tissue.

### 3.4. Time-Controlled Alveolar Recruitment

The TCAV method can recruit lung tissue with low mechanical power by extending inspiratory duration (Figure 2B CPAP Phase). We have shown that there is continual alveolar recruitment during the CPAP Phase without an increase in airway pressure. Thus, significant recruitment occurs without the increases in mechanical power or energy ‘cost’ [106] that are associated with VILI [107,108]. (Appendix A). By comparison, the brief inspiratory time using the ARDSnet method (Figure 2A Duration of Inspiration) does not effectively recruit viscoelastic alveoli or effectively redistribute gas from alveolar ducts into alveoli [84,109]. This might be achieved by increasing airway pressure at the ‘cost’ of increasing mechanical power.

As discussed above, TCAV results in continued alveolar and lung tissue recruitment over time without an increase in airway pressure or mechanical power. In a previous study [81], we applied CPAP at 20, 30, and 40 cmH_2_O for 40 s in rats, and measured recruitment on the gross lung, as well as micro-recruitment of subpleural alveoli, using in vivo microscopy. Figure 5 shows the impact of 40 cmH_2_O CPAP held for 40 s on gross and micro-recruitment. There was rapid initial recruitment in the first 2 s followed by gradual and continual recruitment of alveoli with longer opening time constants occurring over the entire 40 s CPAP application (Appendix A). The newly recruited tissue during the CPAP Phase are stabilized by the very brief Release Phase, which acts as a brake to prevent re-collapse.

## 4. How TCAV Works

### 4.1. Inspiratory and Expiratory Time to First Stabilize and then Gradually Recruit Alveoli

The key component of properly adjusted TCAV is the correct setting of the Release Phase duration, also known as T_Low_. The Release Phase duration is directed by changes in lung physiology, measured as C_RS_. The slope of the expiratory flow-time curve (Slope_EF_) is a reflection of C_RS_ on a breath-by-breath basis (Figure 2B, Gas Flow/Time curve). A healthy patient with a functioning pulmonary surfactant system will have a normal EELV and low lung recoil (Figure 6A, thin spring in chest). As EELV and C_RS_ fall with progressive lung injury, lung recoil will increase and yield more rapid expiratory gas flow (Figure 6B, thick spring in chest). With TCAV, the duration of the release phase is set at 75% of the peak expiratory flow (P_EF_) rather than an arbitrary time. For example, if P_EF_ is 60 L min^−1^, then the termination of expiratory flow (T_EF_) occurs at 45 L min^−1^ [(P_EF_ (60 L min^−1^) × 0.75 = T_EF_ (45 L min^−1^)]. Gradual expiratory flow results in a shallow Slope_EF_ taking a longer time to reach the T_EF_ point (Figure 6A, 0.5 s). The ARDS patient with high lung recoil has a rapid expiratory flow which reduces the duration of the Release Phase (Figure 6B, 0.3 s). The longer release phase for the normal lung results in more exhaled volume and pressure loss, increasing V_T_ and reducing TC-PEEP (Figure 6A). The shorter the Release Phase, the smaller the V_T_ and the higher the TC-PEEP (Figure 6B). Thus, the TCAV method matches V_T_ and TC-PEEP (Figure 2B) to lung pathophysiology, and effectively controls the delivered V_T_ in accordance with changes in C_RS_ (Figure 6B).

Figure 7 summarizes the mechanisms by which the TCAV method stabilizes and recruits lung tissue. The Release Phase duration (T_Low_) immediately stabilizes alveoli (Figure 7A,C green box), eliminating atelectrauma. Once the lung has been stabilized, the extended CPAP Phase duration (T_High_) will gradually reopen alveoli with each breath, eliminating stress concentrators (Figure 7B,C yellow box). The short T_Low_ acts as a brake to prevent the newly opened tissue from re-collapsing. Using inspiratory and expiratory time in addition to airway pressures, the acutely injured lung is pulled from the VILI Vortex (Figure 3).

### 4.2. Inflate-and-Brake Method to Ratchet Open Collapsed Alveoli

We postulate that the progressive and gradual alveolar recruitment during the extended CPAP Phase is augmented by an inflate and break “ratchet” effect with each breath. (Figure 8). A ratchet is defined as a mechanism that permits motion incrementally in one direction while preventing motion in the opposite direction. A classic example is a ratchet wheel that moves the object and a pawl to prevent movement in the opposite direction (Figure 8 Ratchet Example A–C). In the example, the bucket moves up the well shaft a small distance with each pull on the ratchet wheel and does not move back down when the force is released because of the pawl hooks in the cog on the ratchet wheel (Figure 8 Ratchet Example A–C).

The TCAV method uses a similar inflate-and-break method to ratchet open small volumes of lung tissue with each breath. First, rapid inspiratory flow recruits a small volume of collapsed tissue, similar to each pull on the ratchet wheel gradually lifting the bucket up the well (Figure 8A–C Gas Flow Time curve, Inspiration red line). The brief Release Phase does not allow enough time for the lung to depressurize fully and is shorter than the rate of lung collapse (Figure 8A–C Gas Flow/Time curve, Expiration red line). This brief expiratory time acts as a “brake” to prevent the re-collapse of newly recruited lung tissue, similar to the pawl that prevents the bucket from moving down the well. The combination of the extended CPAP Phase with the very brief Release Phase will gradually and progressively recruit the entire lung. (Figure 3). Therefore, *time* is a critical control variable in progressive lung recruitment, since appropriate inspiratory and expiratory durations contribute to both recruiting the lung and preventing re-collapse. We have recently demonstrated the sensitivity of alveolar collapse to expiratory time using a computational model [110]. We have shown that ratcheting open a small volume of lung with each breath is highly effective at stabilizing and reopening collapsed alveoli and has significantly reduced lung histopathology and mortality in clinically applicable animal models. However, it is possible that dependent alveoli with poor compliance in ARDS patients may re-collapse during even a very brief expiratory duration. This could also be true of any level of set PEEP using conventional ventilation methods.

## 5. Physiological Correctness: Scientifically Sound Methods to Set and Adjust the APRV Mode

It is clear that the choice of settings for the volume assist/control mode in the ARDSnet protocol (V_T_ = 6 mL kg^−1^ vs. 12 mL kg^−1^) resulted in significant differences in mortality [27]. In other words, *It’s not the Wand, it’s the Wizard*—It is not the mode but the way the mode is set and adjusted. However, the specific manner by which APRV is managed in various clinical trials is rarely considered in viewpoint, review, or meta-analysis publications [111,112,113,114,115,116,117,118].

The most investigated and comprehensively studied method to set and adjust the APRV mode is the TCAV method, which is based on our understanding of the dynamic alveolar volume change in acutely injured lungs [18,81,82,83,84,85,86,87,88,89,98,99,119,120,121,122,123,124,125]. As discussed above, the acutely injured lung becomes dependent on both time and pressure. Based on this knowledge, TCAV tailors the inspiratory and expiratory durations to recruit and stabilize the alveoli in the injured lung. Other methods to set and adjust the APRV mode may not take into account such personalized physiology, such as the following in Section 5.1, Section 5.2 and Section 5.3:

### 5.1. Adjusting the CPAP Phase Pressure (P_High_) to Maintain a Low V_T_ [126,127]

Like all pressure control modes, TCAV does not set a V_T_ but, rather, is a product of the T_Low_, the CPAP Phase pressure (P_High_), and lung recoil. Using the TCAV method, there is no need to make adjustments to either P_High_ or T_Low_ to maintain a V_T_ of ≤6 mL kg^−1^ since a high V_T_ will never be delivered to a patient with severe ARDS with properly set P_High_. The P_High_ is adjusted to be sufficiently high to result in gradual lung recruitment and is dependent on the severity of lung injury and chest wall compliance. In a patient with low C_RS_, the lung will mechanically determine its own “protective” level of V_T_ directed by changes in the Slope_EF_ curve, often to a much lower level than 6 mL kg^−1^. The only way that the V_T_ can increase is if C_RS_ also increases (High C_RS_ = Larger V_T_. (Figure 6A) and Low C_RS_ = Small V_T_ (Figure 6B). As V_T_ increases with improving C_RS_, the ∆P will actually decrease since ∆P = V_T_/C_RS_ because the C_RS_ will increase proportionately more than the V_T_. By dropping P_High_ to maintain a low V_T_, further lung collapse may ensue and push the lung deeper into the VILI Vortex.

### 5.2. Adjusting T_Low_ to Reduce PaCO_2_ [128]

Increasing T_Low_ would increase V_T_, which would increase CO_2_ removal. However, a larger V_T_ could increase atelectrauma. The priority in TCAV is to first stabilize the lung and, thus, the T_Low_ must be set based on C_RS_. In order to effectively prevent RACE (Figure 4 and Figure 7) [83,84,85,98], PaCO_2_ should be controlled by decreasing the T_High_. Reducing T_High_ will increase the respiratory rate to bulk ventilate, reducing arterial CO_2_ levels.

### 5.3. Increasing the CPAP Phase Time (T_High_) before the Lung Has Been Adequately Recruited

An extended T_High_ (4–6 s) will not result in CO_2_ retention if the lung is mostly inflated, offering a large surface area for gas diffusion. However, if TCAV is applied to an ARDS patient with a large volume of collapsed lung, hypercapnia can develop. PaCO_2_ can be lowered by decreasing the duration of the CPAP Phase duration (T_High_), which will increase minute ventilation and bulk ventilate the lung. This ventilator adjustment will eliminate the need to increase T_Low_ and still maintain normocapnia (see Section 5.2 above).

### 5.4. Concerns for Safety Using the TCAV Method

No ventilator mode is safe if the method used to set and adjust that mode is physiologically unsound, including APRV. The TCAV method is the most extensively studied method to set and adjust APRV, with numerous basic science and clinical publications that have been reviewed and discussed extensively in this paper [18,81,82,83,84,85,86,87,88,89,98,99,119,120,121,122,123,124,125]. Thus, extensive evidence exists identifying the mechanisms of TCAV efficacy by direct visualization of subpleural alveoli in real-time, in clinically applicable animal ARDS models, and in a statistical analysis of patients in the surgical intensive care unit (SICU). A large randomized controlled clinical trial (RCT) has been funded in the United Kingdom to test the efficacy of TCAV as compared with the ARDSnet low V_T_ strategy in ARDS patients that will begin this fall. Most of the safety concerns are unfounded and based on myths and misconceptions concerning the APRV mode [90]. A list of the most common myths that have been extensively reviewed includes the following:

APRV causes barotrauma since ∆P is not controlled. To date, there have been no RCTs showing that APRV causes more barotrauma than other ventilator modes (see Myth #2 [90]).APRV can generate large V_T_ as lung compliance improves. There have been no clinical studies using APRV linking an increase in V_T_ to an increase in mortality. Indeed, as lung compliance improves, a larger V_T_ can be used, and ∆P will actually decrease since ∆P = V_T_/Lung Compliance. It has been shown in multiple studies that a low V_T_ used on a patient with higher lung compliance can actually increase mortality (see Myth #3 [90]).APRV will cause hypercapnia. Over the past 25 years, it has been shown that patients on the APRV mode had lower PaCO_2_ levels than those on conventional mechanical ventilation methods when minute ventilation was matched. This suggests that the volume of CO_2_ removed with each breath is greater with APRV than with conventional modes (see Myth #5 [90]).APRV causes hemodynamic compromise. It is well-known that venous return can be impaired during positive pressure ventilation. There have been no clinical studies indicating that this occurs more with APRV than with any other ventilator mode (see Myth #4 [90]).The complexity of the APRV mode limits its widespread use. This suggests that skills necessary to use other modes of ventilation on the ARDS patient are much simpler to grasp, which is not true. Yes, the skill set is different, but it is not inherently more complex and difficult. APRV has been used successfully on tens of thousands of patients for over 30 years and continues to be used to generate a large volume of empirical data (see Myth #1 [90]).APRV is an Open Lung Approach (OLA) and, thus, is similar to those OLA using conventional ventilation methods that have not shown reduced mortality. Indeed, the whole point of this proof-of-concept manuscript is that TCAV is not a typical OLA, but rather a novel stabilize and then gradually recruit method, as suggested in the title. We review the current OLA strategies in Section 2.2. *Open Lung Approach*. Using the TCAV method, the brief expiratory time will very quickly stabilize alveoli, minimizing RACE-induced atelectrauma, a major VILI mechanism. The lung can then be gradually and safely reopened over an extended period of time (hours or days). Lung reopening is accomplished using the inflation and brake ratcheting open of tissue with each breath, combined with an extended CPAP Phase that recruits alveoli with long-opening time constants. Contrast this with a conventional low V_T_ and PEEP method with recruitment maneuvers designed to open the lung in minutes or seconds, which does not afford durable lung recruitment [69]. It is true that the TCAV method will open the lung, but the method and timing are completely different from the methods used with conventional ventilation methods.

## 6. Conclusions

The current methods of protective mechanical lung ventilation have not substantially reduced the mortality associated with ARDS. TCAV is a personalized and adaptive approach designed to stabilize alveoli rapidly, reduce progressive lung collapse, and avoid a spiral into the VILI Vortex. Although often used as a rescue mode, TCAV is simply CPAP with a release and can be applied as soon as the patient is intubated so the lung never enters the Vortex. After immediately reducing atelectrauma, collapsed lung tissue can be gradually and safely reopened over an extended period of time (hours or days) while simultaneously avoiding excessive dynamic strain or mechanical power. TCAV is the most clinically applied and scientifically studied method to set and adjust the settings of the APRV modality. TCAV results in stable and homogeneous lung inflation, maintains end-expiratory lung volume, minimizes loss of surfactant function, and reduces inflammation and histopathologic markers of lung injury. The mechanisms of TCAV efficacy include adjusting inspiratory and expiratory durations, first to stabilize and then to gradually open collapsed lung tissue and prevent re-collapse. This treatment sequence of stabilization followed by progressive recruitment is a paradigm shift in protective ventilation and is crucial since regional alveolar instability and stress concentrators are the primary mechanisms of VILI. Using such an approach to unshrink the baby lung may be an effective strategy for improved outcomes in ARDS. Of course, there needs to be successful RCT testing TCAV against the current standard of care before this method can become a primary ventilation strategy on a large scale. The key to this RCT is that untested methods to set and adjust APRV cannot be used. TCAV is based on sound physiological principles and is the most studied protocol to set and adjust the APRV mode. As we know, using the volume assist/control mode, a change in tidal volume from 6 cc/kg to 12 cc/kg resulted in a significant mortality increase. The conclusion was not that the volume assist/control mode was ‘bad’, but, rather, that one method to set and adjust the mode was better. After all, It’s not the Wand, It’s the Wizard.

## Figures and Tables

**Figure 1 jcm-12-04633-f001:**
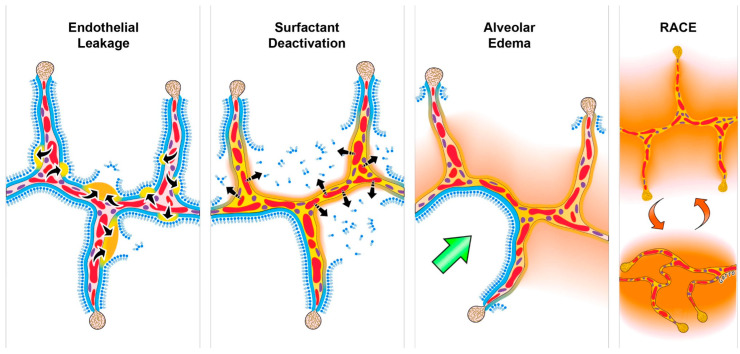
Acute respiratory distress syndrome (ARDS) tetrad of pathologic components [18]. Pulmonary surfactant forms a complete monolayer (blue layer) on the alveolar walls. Pulmonary capillaries (red ovals) run through the alveolar walls. ARDS is a syndrome and can be caused by multiple mechanisms, including severe trauma, hemorrhagic shock, sepsis, and viral or bacterial pneumonia (including that caused by SARS-CoV-2). These injuries initiate systemic inflammatory response syndrome (SIRS), increasing microvascular permeability and causing *Endothelial Leakage* of plasma (black arrows) from the pulmonary capillaries (red ovals). The increased permeability allows pulmonary edema to move into the alveolus (*Endothelial Leakage*—tan edema blebs and black arrows) [19]. As edema expands, the monolayer is disrupted, causing *Surfactant Deactivation* (blue surfactant sluffed into the alveolus). If unchecked, pulmonary edema will eventually flood the entire alveoli (*Alveolar Edema*—tan area within alveolus) [20]. Improperly set mechanical ventilation can exacerbate surfactant disruption [21] initially caused by edema leaking into the alveolus (*Surfactant Deactivation*). High alveolar surface tension can independently increase edema [22], setting up a vicious cycle of increased vascular permeability → alveolar flooding → improper mechanical ventilation → surfactant deactivation → elevated alveolar surface tension → more edema. Alveoli flood in a heterogeneous manner such that there are edema-filled alveoli directly adjacent to air-filled alveoli. The stress of ventilation is concentrated in areas between collapsed or flooded and open alveoli and is known as a stress multiplier. These locations are subjected to stress failure, as alveolar walls are bent toward the edema-filled alveolus (*Alveolar Edema*—green arrow) [23]. *Alveolar Edema* will eventually cause CO_2_ retention and hypoxemia, necessitating the use of mechanical ventilation; however, if inappropriately set, this can cause an unintentional ventilator-induced lung injury (VILI). The mechanisms of VILI include stress multipliers causing volutrauma in adjacent inflated alveoli (green arrow) [23,24,25,26] and unstable alveoli that collapse and reopen with every breath (*Repetitive Alveolar Collapse and Expansion—RACE*), causing atelectrauma. Both volutrauma and atelectrauma exacerbate endothelial leak (*Endothelial Leakage*), accelerating lung damage. Reproduced from Reference [18], under terms of the Creative Commons Attribution 4.0 International License.

**Figure 2 jcm-12-04633-f002:**
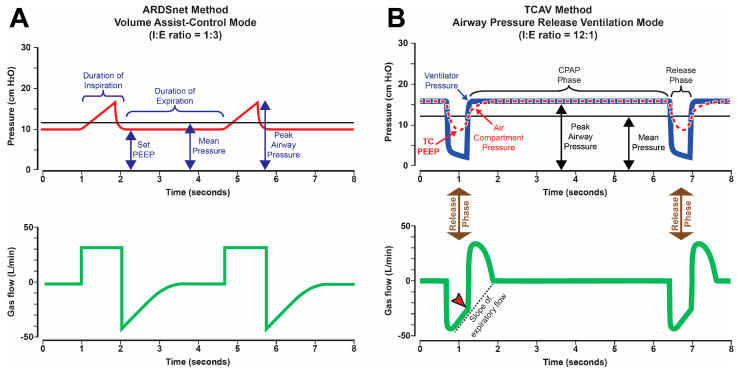
Pressure/Time and Flow/Time curves for the mechanical breath profile (MB_P_) generated by the ARDSnet method to set and adjust the Volume Assist-Control (VAC) and airway pressure release ventilation (APRV) mode, set using the time-controlled adaptive ventilation (TCAV) method. The MB_P_ contains all airway pressures, volumes, flows, rates, and the time they are applied at both inspiration and expiration. (**A**) Pressure/Time and Flow/Time curves for the MB_P_ generated with the VAC mode. Key features of the VAC mode include an inspiratory/expiratory ratio of 1:3, generating a short inspiratory and long expiratory time. There is no extended plateau pressure, so peak inspiratory pressure is very brief. A set positive end-expiratory pressure (Set-PEEP) and FiO_2_ adjusted using oxygenation as the trigger for change [27]. (**B**) Pressure/Time and Flow/Time curves for the MB_P_ generated with the TCAV method to set and adjust the APRV mode. Key features include an inspiratory/expiratory ratio of ~12:1, generating a long inspiratory and short expiratory time. The continuous positive airway pressure (CPAP) Phase is ~90% of each breath. A tidal volume (V_T_), which is measured as the volume of gas released (V_R_) during the Release Phase (brown arrow), is not set but is influenced by changes in, (i) respiratory system compliance (C_RS_), (ii) the CPAP Phase pressure, and (iii) the duration of the Release Phase. The Release Phase is determined by the Slope of the Expiratory Flow Curve (Slope_EF_, red arrow), which is a breath-to-breath measure of C_RS_. The lower the C_RS,_ the faster the lung recoil, the steeper the Slope_EF_, and the shorter the Release Phase, further reducing V_T_. Thus, the V_T_ will always be low in a non-compliant injured lung and will increase in size only when the lung recruits and C_RS_ increases. Since a change in C_RS_ directs the Release Phase duration, which in turn adjusts the V_T_ and the time-controlled PEEP (TC-PEEP), the TCAV method is both *personalized* and *adaptive* as the patient’s lung becomes better or worse: small V_T_ and higher TC-PEEP in the injured lung and larger V_T_ and lower TC-PEEP in the normal lung, always keeping the Driving Pressure (∆P = Vt/C_RS_) in the safe range. Reproduced from Reference [30], under terms of the Creative Commons Attribution 4.0 International License.

**Figure 3 jcm-12-04633-f003:**
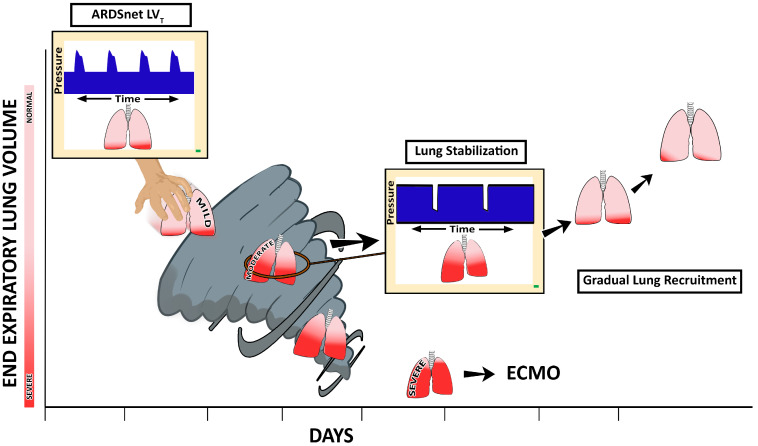
In patients with acute respiratory distress syndrome (ARDS), the evolution of ventilator-induced lung injury (VILI) can be described as an ever-shrinking normal ‘baby’ lung, resulting in a ‘VILI Vortex’. A ventilation strategy that does not prevent progressive lung collapse fuels the VILI Vortex. If unchecked, lung injury will progress into severe ARDS, at which point rescue methods such as extracorporeal membrane oxygenation (ECMO) may be necessary. In order to circumvent this VILI Vortex, methods to quickly stabilize and then gradually reopen collapsed lung tissue must be developed. We hypothesize that a stabilize the lung approach (SLA) that first stabilizes alveoli and then gradually reopens collapsed tissue can be accomplished using the time-controlled adaptive ventilation (TCAV) method to set and adjust the airway pressure release ventilation (APRV) mode. If our hypothesis is correct, this will be a paradigm shift in the way medicine is practiced from open-the-lung *first* and stabilize-the-lung *second* using the Open Lung Approach (OLA) to reversing this order of treatment (*stabilize and then gradually recruit*) using the Stabilize Lung Approach. Reproduced from Reference [64], under terms of the Creative Commons Attribution 4.0 International License.

**Figure 4 jcm-12-04633-f004:**
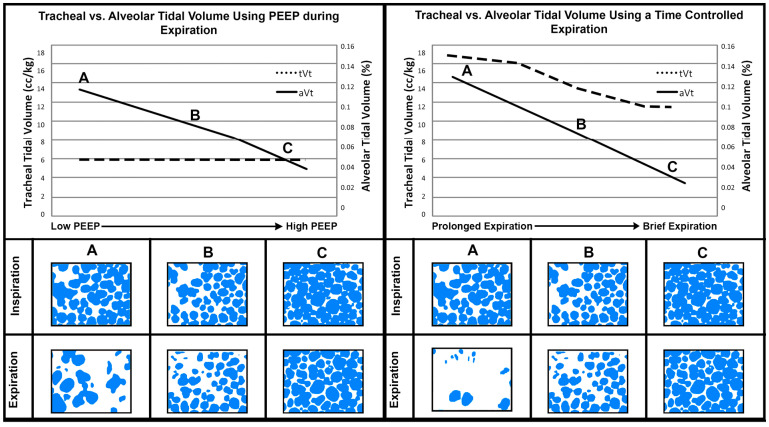
The effect of tracheal tidal volume (tV_T_ solid line, measured by the ventilator) and alveolar tidal volume (aV_T_ dashed line, measured using in vivo microscopy of subpleural alveoli, as the difference in alveolar size at inspiration and expiration) with increasing PEEP or decreasing Expiratory Duration (A → C). Low PEEP or a Prolonged Expiration (A) result in alveolar collapse (fewer open blue alveoli during expiration) and fewer alveoli open during inspiration. Applying High PEEP (C) increases the number of open alveoli available to accommodate the tidal volume delivered by the ventilator and, thus, even though the tV_T_ remains constant (6 cc/kg—dashed line), the aV_T_ decreases (~0.12% to 0.04%—solid line). As Expiratory Duration is reduced, both tV_T_—dashed line—and aV_T_—solid line—decrease. In addition, more alveoli remain open at end-expiration (C—blue subpleural alveoli). Using a Brief Expiration, even the presence of tV_T_~12 cc/kg (C—dashed line) results in a lower aV_T_ as compared with that with High PEEP (C, ~0.04% vs. 0.02%—solid lines). Subpleural alveoli were filmed using in vivo microscopy and color-coded blue using computer image analysis. Alveolar tidal volume was measured as the percent change (%) in the area of the photomicrograph occupied by alveoli (blue) from inspiration to expiration. With Permission [105].

**Figure 5 jcm-12-04633-f005:**
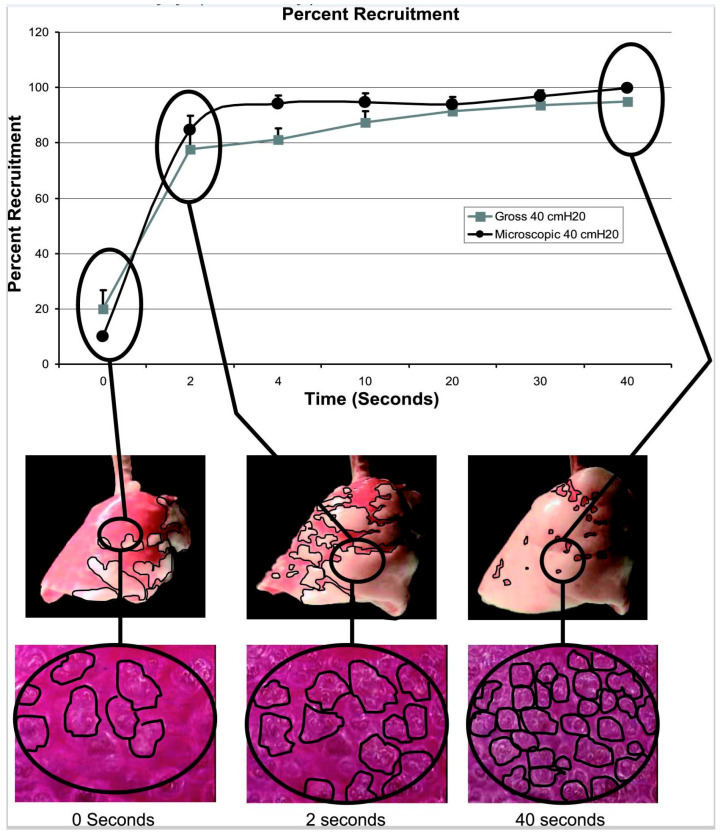
Lung tissue and alveolar recruitment over time without an increase in airway pressure. An airway pressure of 40 cmH_2_O was applied at T0 and held for 40 s (T40). Bottom panel shows subpleural alveoli (black circles), middle panel shows lung tissue (dark red = atelectasis; pink = inflated tissue) at T0, T2, and T40 after airway pressure was applied. Top panel is the Percent Recruitment/Time curve with alveolar recruitment measures as the percent of the photomicrograph occupied by inflated alveoli and, for the whole lung, the percent of the entire lung surface occupied by inflated (pink) tissue. There was a slight pause following the application of airway pressure followed by rapid alveolar recruitment by T2. The lung continued to recruit over 40 s without an increase in airway pressure (Appendix A). With Permission [81].

**Figure 6 jcm-12-04633-f006:**
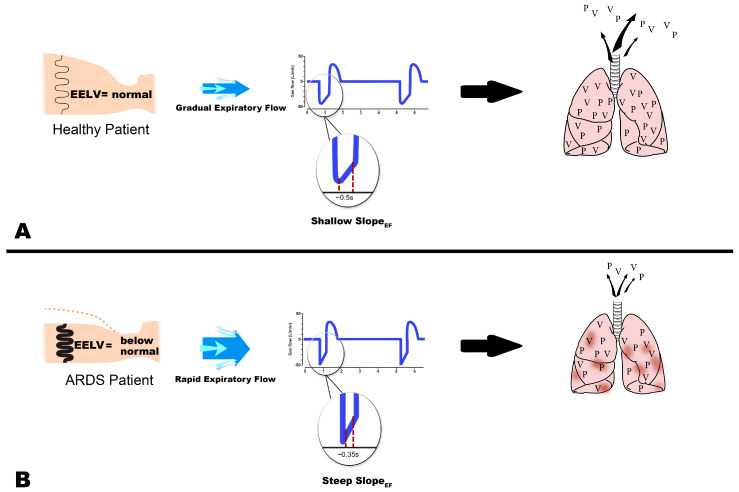
The time−controlled adaptive ventilation (TCAV) method to set and adjust the airway pressure release ventilation (APRV) mode delivers a personalized tidal volume (V_T_) and time-controlled positive end-expiratory pressure (TC-PEEP) directed by lung physiology (Figure 2B, TCAV Method). (**A**) In a healthy patient, lung recoil is small (thin spring), resulting in slow expiratory gas flow. This can be identified on the ventilator monitor as a Shallow Slope of the expiratory flow curve (Slope_EF_). The slow gas flow results in a relatively extended expiratory duration because it takes longer to reach the termination of expiratory flow (T_EF_) point (0.5 s). Using the TCAV method T_EF_ is determined by multiplying the peak expiratory flow (P_EF_) by 75% (P_EF_ L/min × 75% = T_EF_ L/min). In a healthy patient, the V_T_ is larger (many ‘V’ leaving the lung) and the lung has a longer time to depressurize (many ‘P’ leaving the lung), lowering the TC-PEEP, because of the longer expiratory time. (**B**) Lung collapse, pulmonary edema, and surfactant deactivation in an ARDS patient’s lung greatly increase lung recoil (thick spring), causing a rapid expiratory gas flow seen as a steep Slope_EF_ on the ventilator monitor. The rapid expiratory flow results in the T_EF_ being reached very quickly (0.35 s). The very brief expiratory duration allows only a small volume of gas to be exhaled. This significantly reduces V_T_ (few ‘V’ leaving the lung) and does not allow time for the lung to depressurize (few ‘P’ leaving the lung), maintaining a higher TC-PEEP and end-expiratory lung volume (EELV). Using the TCAV method, the expiratory duration (T_Low_) determines both the V_T_ and TC-PEEP levels, which are personalized and adaptive as the lung becomes better or worse. ‘V’ = Volume; ‘P’ = Pressure.

**Figure 7 jcm-12-04633-f007:**
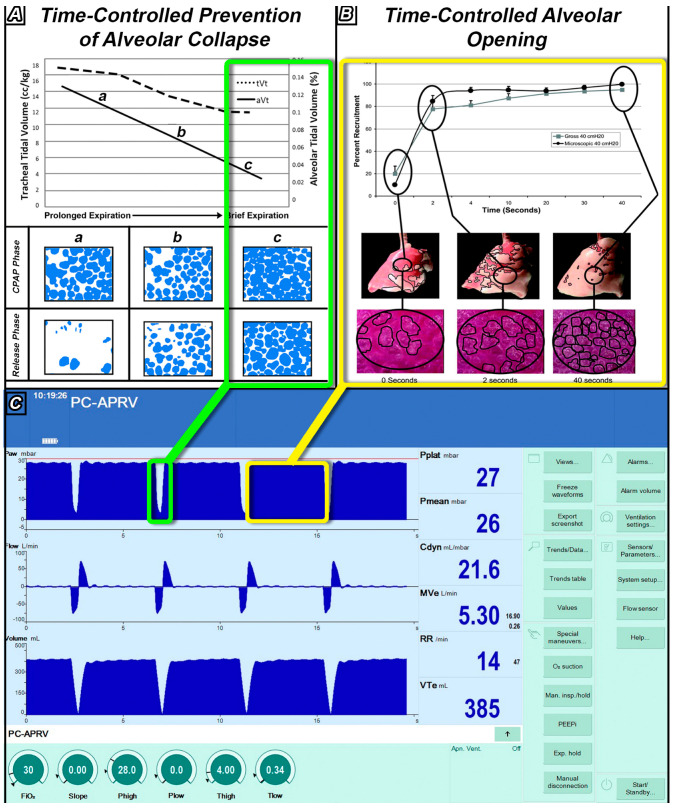
The impact of expiratory and inspiratory time on alveolar collapse and recruitment. ((**A**) **Top**) A graph of the tracheal tidal volume (tV_T_, dotted line) measured by the ventilator, and subpleural alveolar tidal volume (aV_T_, solid line) measured using in vivo microscopy at three Release Phase durations: (a) Prolonged, (b) Moderate, and (c) Very Brief. Both tV_T_ and aV_T_ decrease as the Release Phase is shortened. Alveolar tidal volume is very small (c 0.02%—solid line) with a very brief Release Phase (green box), even with a large tV_T_ (12 cc/kg—dashed line) [105]. A large alveolar tidal volume (aV_T_—solid line) is a direct measurement of repetitive alveolar collapse and expansion (RACE). ((**A**) **Bottom**) Subpleural alveoli (Blue circles) measured using in vivo microscopy during the CPAP Phase and Release Phase using the APRV mode. With a prolonged Release Phase (a, b), alveoli collapsed and did not all reinflate during the CPAP Phase (large white areas between blue alveoli). With a brief Release Phase set by the TCAV method (c—green box), alveoli were open and stable (i.e., no RACE) throughout the ventilatory cycle (green box, blue alveoli fill the photomicrographs at both the CPAP and Release phases). With Permission [105]. ((**B**) **Top**) A graph showing whole lung and alveolar recruitment over time with static applied pressure. The Percent Recruitment/Time curve following 40 cmH_2_O airway pressure on both subpleural alveoli (black line) and the gross lung surface (gray line). Following the application of airway pressure, there was a very slight delay in opening, followed by short rapid recruitment, and then continual recruitment as long as the airway pressure was applied. With Permission [81]. ((**B**) **bottom**) In vivo microscopy showing progressive subpleural alveoli recruitment (circled in black) (Appendix A) and the progressive recruitment of lung tissue (red tissue turning pink) (Appendix A) over 40 s at a constant airway pressure. Lung tissue and alveoli demonstrate viscoelastic behavior and continue to recruit throughout the CPAP Phase without an increase in airway pressure [81]. With an extended CPAP Phase set by the TCAV method (yellow box), alveoli (black circles) and lung tissue (red tissue turning pink) are gradually and continually recruited over 40 s without an increase in airway pressure (green box). (**C**) Ventilator monitor showing Pressure/ Flow and Volume/Time curves set using the TCAV method. The very short Release Phase stabilizes alveoli, preventing collapse (green boxes). The extended CPAP Phase recruits alveoli for the entire time the pressure is applied (yellow boxes). Subpleural alveoli were filmed using in vivo microscopy and color-coded blue using computer image analysis. Alveolar tidal volume was measured as the percent change (%) in the area of the photomicrograph occupied by alveoli (blue) from inspiration to expiration [105]. Percent Recruitment for alveoli was measured as the percent of the photomicrograph occupied by inflated alveoli and, for the whole lung, the percent of the entire lung surface occupied by inflated (pink) tissue.

**Figure 8 jcm-12-04633-f008:**
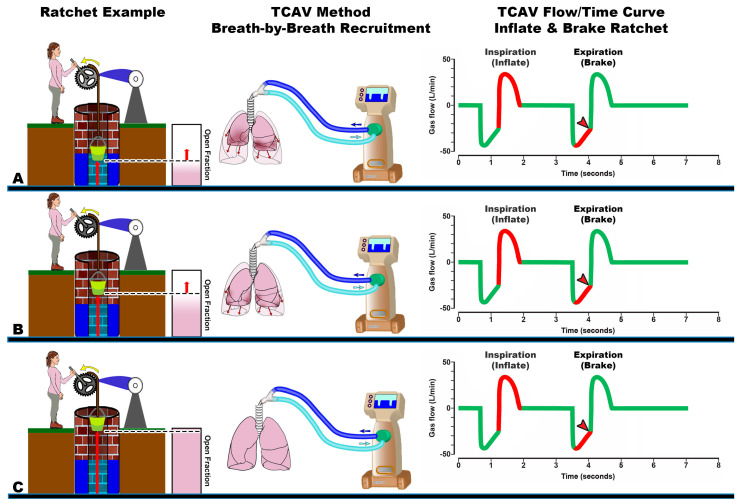
Ratchet concept of breath-by-breath gradual lung recruitment. A ratchet is a mechanical device (ventilator) that allows a continuous linear motion in only one direction (lung inflation), while preventing motion in the opposite direction (lung re-collapse). *Ratchet example*: (**A**−**C**) An individual pulls the lever and turns the ratchet wheel (yellow arrow) to progressively lift the bucket with each stroke up the well shaft (Open Fraction, red arrow). The pawl acts as a brake to prevent the bucket from moving back into the well. *TCAV Method Breath-by-Breath Recruitment*: (**A**) When TCAV is first applied, the end-expiration lung volume (EELV) is below normal, depicted as the densely colored tissue within the fully inflated lung shadow. Areas of atelectasis are shown as heterogeneous dark regions on the lung surface (Figure 6, ARDS Patient, EELV below normal). (**B**) With each breath, a small amount of tissue is opened, increasing the Open Fraction (Red arrow) until the lung is fully recruited (**C**). *TCAV Flow/Time Curve Inflate and Brake Ratchet*: The TCAV Gas Flow/Time Curve (Figure 2B). (**A**−**C**) Inspiration—Rapid lung inflation (red line) inflates a small portion of lung tissue with each breath. Expiration—The very brief expiratory duration (red line and arrowhead) acts as a brake to keep the newly open lung tissue from re-collapsing. Together, this inflate and brake system will progressively ratchet open lung tissue, over hours or days.

**Table 1 jcm-12-04633-t001:** Problems with the low tidal volume (LVt) Protective Lung Approach (PLA) [27]. Although protecting normal tissue from over-distension and ‘resting’ the collapsed lung sounds protective, the lung is a dynamic organ designed to inflate and deflate continually. Long-term atelectasis (resting) generates multiple pathophysiologic problems including.

***Problem 1*:** Reducing V_T_ in patients with higher compliance can increase mortality [29].
***Problem 2***: Using LV_T_ and airway pressure results in atelectasis-induced loss of lung volume, resulting in more stress and strain applied to the remaining normal tissue during ventilation (‘VILI Vortex’) [13].
***Problem 3***: Areas of atelectasis that remain collapsed, known as collapse induration, can become permanently dysfunctional and fibrotic [50,51,52,53].
***Problem 4***: Atelectasis shrinks perfusion surface area, causing hypercapnia and hypoxemia resulting in higher FiO_2_ requirements, which in turn can cause oxygen toxicity and adsorption atelectasis, exacerbating all problems associated with lung collapse. Also, atelectasis is independently associated with loss of surfactant function and carries an increased risk of pneumonia [21,54].
***Problem 5***: Hypoxemia, hypercapnia, and stretch receptors in the atelectatic tissue cause dyspnea, resulting in patient–ventilator asynchrony [55,56].
***Problem 6:*** To prevent repetitive alveolar collapse and expansion (RACE)-induced atelectrauma, the LV_T_ approach uses changes in oxygenation to guide the settings and adjustments of PEEP. However, oxygenation is a very poor indicator of alveolar instability [57,58,59].
***Problem 7*:** End-expiratory lung volume (EELV) decreases as atelectasis increases, which increases the pulmonary vascular resistance (PVR) [60]; this can lead to right heart failure requiring vasoactive therapy [61].
***Problem 8:*** Low V_T_ and respiratory rate decrease lung lymph, which could exacerbate pulmonary edema accumulation [62].
***Problem 9:*** Lung stretch during ventilation is needed to stimulate exogenous surfactant release [63].

## Data Availability

Not applicable.

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
