# Peer review of "First Stabilize and then Gradually Recruit: A Paradigm Shift in Protective Mechanical Ventilation for Acute Lung Injury"

_jcm, 2023, doi:10.3390/jcm12144633_

Round 1

Reviewer 1 Report (Previous Reviewer 1)

Thank you for submitting the revision.

The article is well written and the ideas/concept well defended. I am suggesting to the editor to show as 'proof of concept' or 'point of view' pending stronger evidence.

A minor point: do you think spontaneous breathing early (during stabilization phase) can affect recruitment/cycling/barotrauma or you think it can be of benefit (e.g., haemodynamics). I understand that the article is not about APRV, but this may a point to propose for future research when proving the concept.

Thank you

Author Response

As long as spontaneous breathing is a rhythmic normal cycle it is fine and helpful. If the spontaneous breath is dyssynchronous then it can be harmful. Using the TCAV method dyssynchrony only occurs when it is first applied before the lung is gradually reopened. Once the lung is open dyssynchrony is eliminated because of the drive to increase ventilation (low PO2, high CO2, and stretch receptors in the collapsed lung) have been eliminated once the lung is reopened. The length of the CPAP Phase can be reduced to increase minute ventilation that will improve both blood O2 and CO2 concentrations that drive dyssynchrony. It is well known that dyssynchrony does not often occur using CPAP and TCAV is simply CPAP with a release. The patient can inhale or exhale during the CPAP Phase with little resistance also reducing dyssynchrony.

Reviewer 2 Report (New Reviewer)

The manuscript with the title " First Stabilize and then Gradually Recruit: A Paradigm Shift in  Protective Mechanical Ventilation for Acute Lung Injury" has been reviewed by us now. In this review paper, the authors examined the alternative strategies to ventilating the injured lung. The topic of the paper is original and relevant in the field. It addresses a specific gap in the field. They suggest a method that adjusts inspiratory and expiratory durations and inflation pressure levels. A personalized and adaptive approach designed to stabilize alveoli rapidly can reduce progressive lung collapse. However it should be tested more and more patients and centers. The text is clear and compatible with the literature. The paper presents an innovation and contributes to the literature. Also, the design and the writing of the paper are also convenient and fine. Thanks the authors for their labour-intensive review manuscript.

Author Response

We agree that more clinical studies are necessary to move the TCAV method into a primary protective ventilation strategy worldwide. To this end, our group has been funded to conduct a prospective RCT testing the TCAV method vs the standard of care low VT in 40 hospitals on patients with established ARDS.

Reviewer 3 Report (New Reviewer)

Thank you very much for the opportunity to review this informative review article on the TCAV approach to setting APRV for the purposes of, in the authors' words, "stabilizing and then gradually recruiting" the lung in ARDS.  The authors have done a lot of work on this subject and so much of this review is a synopsis of that, which is OK in my opinion as long as the tone and content is neutral and dispassionate rather than partisan, which would not be appropriate for a review.  There is a lot of polarization in the critical care world surrounding APRV/TCAV, so the subject matter of this review is a bit controversial but nonetheless deserving of representation in the literature.  Also, there are some nice figures included in this manuscript (unfortunately I was unable to access the videos).  For the purposes of my review, I am going to give the authors the benefit of the doubt when it comes to the soundness of their physiological arguments to the extent that they are supported at least by animal experiments (some of them their own), so my comments will focus on general issues addressing which could improve the quality of the manuscript:

1. I would define "strain" for the reader when it comes to lung inflation since it's an important concept referred to throughout the manuscript.  

2. I would refrain from citing references in the figure legends (aside from referring to the original source of the image).

3. P4 L156: there is something strange with the reference numbering there.

4. The acronym RACE is defined in a figure legend but never in the body of the text.  It should be expanded in the text itself.

5. The paragraph in section 2.1 is quite repetitive with the paragraph on the same page in lines 142-150.  Perhaps the two can be condensed into one.

6. P5 L186: "which  the vast majority of studies suggest is not the case" this clause is confusing, creates double-negatives in the sentence so I recommend revising.

7. P5 L199: "latter hypothesis" - which one is the "latter" hypothesis?

8. I had a problem with the legend for Fig 3.  The figure itself and the message it sends are fine, but there is too much editorializing by the authors in the legend, which is also disproportionately long.  Please rewrite the legend ion a more neutral tone just describing the significance of the figure rather than pitching your hypothesis.

9. P7 L240: "natural" is duplicated.

10. "By combining TC-PEEP and a very brief expiratory duration..." Isn't TC-PEEP created by the presence of a very brief expiratory duration--if so, then this sounds redundant.

11. Figure 7 panel B appears to be a repeat of Figure 5.

12. The manuscript starts to become problematic with section 5.  In that section, the authors appear to pivot to almost a rebuttal of the viewpoint article published by Kuljit et al in a different journal.  In much of that section, they engage in a passionate defense of the TCAV method that would be appropriate for a pro/con or perspective article type but is out of place in a plain review of the subject.  I think that as a pre-requisite for publication the authors need to rewrite parts of that section in a more neutral tone--the section could summarize the advantages and potential disadvantages of TCAV, but it shouldn't read as though the authors are engaging in a direct debate with a particular author or community within CCM.    

Author Response

  1. I would define "strain" for the reader when it comes to lung inflation since it's an important concept referred to throughout the manuscript.  

Response 3.1: “Strain” has been defined in the text on line 101.

  1. I would refrain from citing references in the figure legends (aside from referring to the original source of the image).

Response 3.2: I feel the references in the legends are important to make each figure a stand-alone piece of information independent from the text. I think this is key since the concepts we are discussing are novel so it is important to have each figure not only illustrating a key point but the reference on hand in the legend for the reader to quickly analyze the science supporting these concepts.

  1. P4 L156: there is something strange with the reference numbering there.

Response 3.3: The references have been realigned. 

  1. The acronym RACE is defined in a figure legend but never in the body of the text.  It should be expanded in the text itself.

Response 3.4: RACE is defined in the Abstract Line 17 and again in the text on Line 159.

  1. The paragraph in section 2.1 is quite repetitive with the paragraph on the same page in lines 142-150.  Perhaps the two can be condensed into one.

Response 3.5: The text in Line 142-150 discusses the physiologic importance of lung compliance in the form of driving pressure as a more robust indicator than tidal volume alone in stratifying ARDS-related mortality risk with a focus on the ARDSnet 1,096 patient database. The 2.1 section paragraph introduces the Protect the lung Approach and the major problems with this method. One of the problems is that it is a one size fits all using a fixed tidal volume of 6cc/kg and not set and adjusted by driving pressure. Thus, we do briefly mention this fact that was discussed in detail in lines 142-150. Other than that, these are very different paragraphs and we feel both are necessary to make our discussion clear to the readers.

  1. P5 L186: "which  the vast majority of studies suggest is not the case" this clause is confusing, creates double-negatives in the sentence so I recommend revising.

Response 3.6: Thank you for pointing this out, the sentence has been revised.

  1. P5 L199: "latter hypothesis" - which one is the "latter" hypothesis?

Response 3.7: This sentence has been modified for clarity.

  1. I had a problem with the legend for Fig 3.  The figure itself and the message it sends are fine, but there is too much editorializing by the authors in the legend, which is also disproportionately long.  Please rewrite the legend ion a more neutral tone just describing the significance of the figure rather than pitching your hypothesis.

Response 3.8: The legend has been rewritten to be more neutral.

  1. P7 L240: "natural" is duplicated.

Response 3.9: One of the ‘natural’ has been removed

  1. "By combining TC-PEEP and a very brief expiratory duration..." Isn't TC-PEEP created by the presence of a very brief expiratory duration--if so, then this sounds redundant.

Response 3.10: Good point, the sentence has been modified.

  1. Figure 7 panel B appears to be a repeat of Figure 5.

Response 3.11: This was done intentionally. Figure 5 discusses the details of the original basic science study showing that alveoli continually recruit with an applied airway pressure over time without adding any additional airway pressure. The panel from the original study is used in Figure 7 to show the reader how the extended CPAP Phase can continue to recruit alveoli based on the original data shown in Figure 5

  1. The manuscript starts to become problematic with section 5.  In that section, the authors appear to pivot to almost a rebuttal of the viewpoint article published by Kuljit et al in a different journal.  In much of that section, they engage in a passionate defense of the TCAV method that would be appropriate for a pro/con or perspective article type but is out of place in a plain review of the subject.  I think that as a pre-requisite for publication the authors need to rewrite parts of that section in a more neutral tone--the section could summarize the advantages and potential disadvantages of TCAV, but it shouldn't read as though the authors are engaging in a direct debate with a particular author or community within CCM.    

Response 3.12: The paragraph has been written in a more neutral tone.

Reviewer 4 Report (New Reviewer)

I'd like to offer my congratulations to the authors on this excellent work. I have a few comments they might consider:

1. the rationale and physiology of TCAV is very well described, however, the implementation of this strategy is somewhat lacking. I'd like to have the views of the authors on how these ventilation modes can be made part of normal practice in ICUs where nursing and medical staff might be less than abundant. My only safety concern with TCAV/APRV is that an in-depth understanding of the potential issues and their solutions is needed at 2 am in the morning at the bedside, which might not be achievable in many units. What is the proposed solution for this?

2. The authors position TCAV as a personalised ventilation approach. This does make physiological sense, but again a recent RCT failed to show benefit of personalised ventilation settings vs standard guideline driven care. Is there a big enough outcome benefit, which would offset any harm of "tinkering" with the ventilator?

Author Response

  1. the rationale and physiology of TCAV is very well described, however, the implementation of this strategy is somewhat lacking. I'd like to have the views of the authors on how these ventilation modes can be made part of normal practice in ICUs where nursing and medical staff might be less than abundant. My only safety concern with TCAV/APRV is that an in-depth understanding of the potential issues and their solutions is needed at 2 am in the morning at the bedside, which might not be achievable in many units. What is the proposed solution for this?

Response 4.1: This is a major problem in the movement of the TCAV method into primary protective ventilation strategies worldwide. To this end, we have a website with the TCAV protocols (TCAVnetwork.org) but personal instruction is necessary. We do offer hands-on Workshops but these have limited reach. We have found that the most effective method of adequate training is for a clinician to work in our lab or with us in the clinic for 6 months or more and then they can train their faculty and staff in their ICU so everyone knows what to do at 2 am. There are several of our trainees who have incorporated TCAV as a primary method of ventilation in their ICUs,

  1. The authors position TCAV as a personalised ventilation approach. This does make physiological sense, but again a recent RCT failed to show benefit of personalised ventilation settings vs standard guideline driven care. Is there a big enough outcome benefit, which would offset any harm of "tinkering" with the ventilator?

Response 4.2: Using the TCAV method we are not ‘tinkering’ with the lung but rather allowing the lung to ‘decide’ the length of the Release Phase. Setting the Release Phase correctly is the key component in setting TCAV correctly since this will immediately stabilize the alveoli pulling the lung from the VILI Vortex. The Release Phase is set and directed by respiratory system compliance (CRS) measured as the slope of the expiratory flow curve, which is a breath-by-breath measure of CRS. As CRS gets worse the Release Phase becomes shorter and vice versa. The shorter Release Phase as CRS falls will prevent alveoli, even with very fast collapse time constants, from de-recruiting during exhalation.

This manuscript is a resubmission of an earlier submission. The following is a list of the peer review reports and author responses from that submission.

Round 1

Reviewer 1 Report

Thank you for drafting this comprehensive review. The concept is very clear and the physiological rationale very well explained.

I admit enjoying reading the paper, but despite many meta-analyses there is still some debate around ‘patient-centered’ outcome, namely the mortality. I would suggest it will be useful if you add 2-3 sentences about what you recommend for future RCTs.

Minor comments:

L 171: 6c ml – c needs to be removed

Table 1 problem 4: The negative impact of high FiO2 may exceed just adsorption atelectasis.

L218: developed or designed

L296: ‘Such lung protection was achieved even though the APRV group had a significantly higher VT compared to the conventional group’ and L494 ‘properly set PHigh’ – how you suggest the Phigh to be set if the release (tidal) volume is not a target?

L498: ‘As VT increases with improving CRS the P will actually decrease since P = VT/CRS.’ – the statement is slightly confusing to me: If the increase in compliance and tidal volume are proportional, delta P should remain constant, except if the intrinsic PEEP changes as well. Can you explain please?

L501: ‘Adjusting TLow to reduce PaCO2’ – Do authors advocate allowing a permissive hypercapnia? In this case, what will be the accepted limits for PaCO2/pH?

Some ventilators introduced a synchronised switching between P high and P low, do you think this is of benefit or can interfere with a proper TCAV? I appreciate you may find this point out of the scope of the article, so you can ignore if you want.

Do authors think there is a value of measuring/calculating the driving pressure during APRV; or the TCAV personalised settings is enough?

Unfortunately, APRV/TCAV are commonly used as rescue mode after hours or sometimes days of conventional ventilation. It is clear that the paper advocates early TCAV APRV, but this was not mentioned in a plain sentence (may be in the conclusion).

Thank you very much
